# LOW ENTROPY DEEP NETWORKS

## ABSTRACT

The movement of data between processor and memory, not arithmetic operations, dominates the energy cost of inference computations in deep networks. Network compression offers opportunities for hardware design to bring weights in memory closer to processor reducing these data movement costs. To this end, we investigate the merits of a method, which we call Weight Fixing Networks (WFN). We design the approach to realise four model outcome objectives: i) very few unique weights, ii) low-entropy weight encodings, iii) unique weight values which are amenable to energy-saving versions of hardware multiplication, and iv) lossless task-performance. Some of these goals are conflicting. To best balance these conflicts, we combine a few novel (and some well-trodden) tricks; a novel regularisation term, (i, ii) a view of clustering cost as relative distance change (i, ii, iv), and a focus on whole-network re-use of weights (i, iii). Our Imagenet experiments demonstrate lossless compression using 56x fewer unique weights and a 1.9x lower weight-space entropy than SOTA quantisation approaches.

## 1 INTRODUCTION

**The Importance of Data Movement Costs.** Although there has been a significant amount of attention exploring both algorithmic (Sze et al., 2020) and hardware-based (Chen et al., 2020) approaches to reducing the energy costs of deep learning inference, there is often a noted disconnect between the two (Sze et al., 2017). The most expensive energy costs lie in memory reads (Horowitz, 2014; Gao et al., 2017). For every off-chip DRAM data read, you pay the equivalent of over two hundred 32-bit multiplications in energy costs[1] (Horowitz, 2014). Algorithmic techniques that hope to reduce energy consumption have focussed predominantly on metrics like floating-point operations (FLOPs); intuitively, fewer FLOPs should translate into smaller energy costs from the reduced number of multiplications and potentially, fewer parameters to read from memory. However, it has been observed that this is a weak proxy for the energy consumed in running a deep learning model for inference (Sze et al., 2017). Data movement costs can still be high if we do not consider the re-use of weights, the delay between their re-use, and filter shape effects. Simply reducing the FLOPs alone does not guarantee energy cost reductions – particularly those incurred through data read and write activity.

**Accelerator Design Considerations.** These data movement costs are difficult to reduce in von Neumann architectures (Li et al., 2015; Sebastian et al., 2020), and so there is a move towards co-design of algorithm-hardware inference accelerators. A core consideration of accelerator design lies in exploring dataflow mappings. These mappings determine how data used in computation is distributed across memory components and optimised to take advantage of the re-use of weights, matrix multiplication partial-sums (psums), and data inputs (Han et al., 2016; Chen et al., 2017; 2015; 2020). Modern deep learning accelerators also make use of hundreds of processing elements (Chen et al., 2020) (PE's) for computation. Each PE contains a small amount of cheap and fast access memory to store a few pieces of information (weights, psums, inputs, etc.). A dataflow mapping cuts down data movement costs by making the best use of each PE and ensuring that the information they contain is recycled as much as possible. A weight/input/psum that cannot be used quickly after its latest use will be written to storage and require a re-read later. Less re-use leads to increased data movement costs. Consider, for example, weight-stationary dataflow, commonly used in SOTA accelerator designs (Farshchi et al., 2019; Jouppi et al., 2017). Here the items stored

---

[1] 45nm process

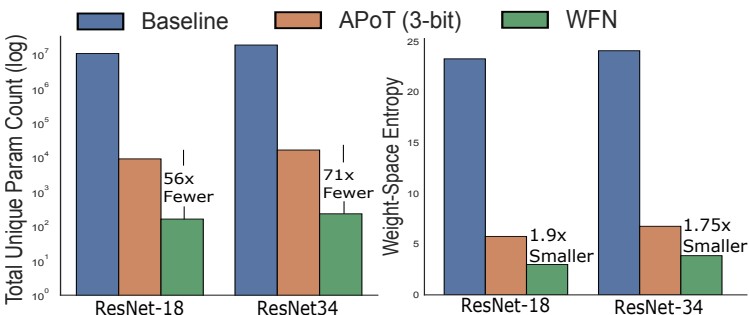

Figure 1: WFN has more weight re-use opportunities than existing quantisation approaches which can be used to reduce data movement costs. **Left**: The total number of unique parameters left after quantisation is 56x fewer than APoT for ResNet-18 trained on the Imagenet dataset and 71x for the ResNet-34 model. **Right**: The entropy of the parameters across the network is 1.9x and 1.65x smaller when using the WFN approach over APoT.

statically in the PE are the model weights. Input data is then fed into the relevant PE's where multiplication is conducted locally. The ideal situation for a dataflow mapping would be to pay a single data movement cost for each unique weight value and then reference these weight values using indexing. The indexing costs can be kept low using an appropriate encoding scheme such as Huffman coding and approximated by the entropy of the weight space. Thus, the indexing access costs plus unique value access costs can be much smaller than the unquantised network (Mao & Dally, 2016; Wu et al., 2018).

**Objectives.** So we ask ourselves what we could do algorithmically to maximise the benefit of accelerator dataflows? We think an easy win is to reduce the number of *unique* weights a network uses. Fewer *unique* weights whilst fixing the network topology and the total number of parameters will mean that more weights are re-used more often. This additional re-use gives more opportunity to dataflows to maintain often-used weights in PE's. To further enhance the compressibility, it is desirable for the distribution of the unique weights to be concentrated around a handful of values. The high probability density weights would then be used more often and could then be reliably stored inside PE's, saving both the cost of overwriting these weight values and re-fetching them when needed later. Finally, we ask what the ideal values of these weights would be. From a computational perspective, not all multiplications are created equal. Integer powers-of-two, for example, can be implemented as simple bit-shifts. Mapping the weights used most to these values offers potential further energy reductions. Putting these three requirements together: few unique weights; a low-entropy encoding with a distribution of weights highly concentrated around a tiny subset of values; and a focus on powers-of-two values for weights — all motivated to reduce computation costs in accelerator designs — we present our contribution.

**Weight Fixing Networks.** Our work's overarching objective is to transform a network comprising many weights of any value (limited only by value precision) to one with the same number of weights but just a few unique values. Rather than selecting the unique weights a priori, we let the optimisation guide the process in an iterative *cluster-then-train* approach. In each iteration, we cluster an ever-increasing subset of weights to one of a few cluster centroids. We map the pre-trained network weights to these cluster centroids, which are the pool of unique weights. The training stage follows standard gradient descent optimisation to minimise performance loss with two key additions. Firstly, only an ever decreasing subset of the weights are *free* to be updated. And secondly, we use a new regularisation term to penalise weights in proportion to their nearest clusters' relative distance. We iteratively cluster subsets of weights to their nearest cluster centre, as has been demonstrated successfully previously (Zhou et al., 2017). The way we determine which subset to move is a core component of our contribution which leads to the superior compression results we achieve.

**Small Relative Distance Change.** Rather than selecting subsets with small Euclidean distances to cluster centres, or those that have small magnitude (Zhou et al., 2017), we make the simple

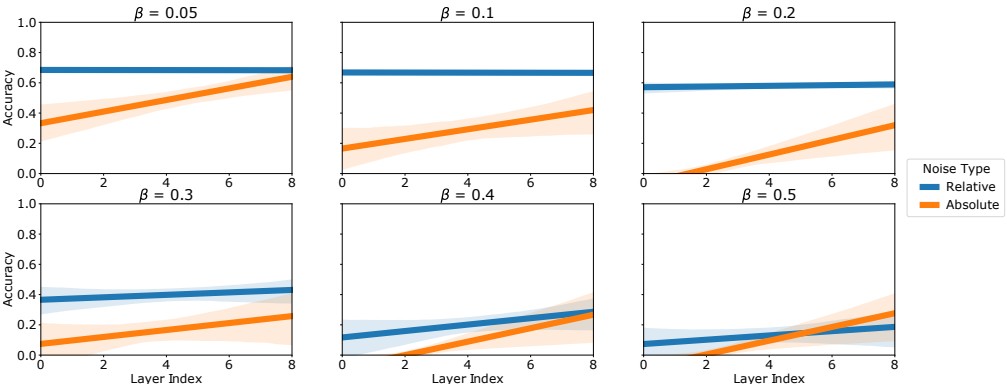

Figure 2: We explore adding relative vs absolute noise to each of the layers (x-axis). The layer index indicates which layer was selected to have noise added. Each layer index is a separate experiment with the 95% confidence intervals shaded.

observation that the *relative* – as opposed to absolute – weight change matters. The distance a weight is moved when quantised is dependent on the distance between the weight $w_i$ and its new value $w_i + \delta w_i$. When the new value is zero — as is the case for pruning methods — then the magnitude of the weight *is* the distance. Yet, this is not the case more generally. We demonstrate the importance of optimising quantisation for small relative changes with simple empirical observations. Using a pre-trained ResNet-18 model, we test adding relative vs absolute noise to the layers' weights and measure the accuracy change. For relative noise we set a noise level $\beta$ and adjust all weights $w_i$ in a layer $l$ as: $w_i^l \leftarrow w_i^l + \mathcal{N}(0, \beta|w_i^l|)$. The standard deviation of noise added is determined by the original value $w_i^l$. We contrast this with absolute noise experiments, where we instead set $w_i^l \leftarrow w_i^l + \mathcal{N}(0, \beta\overline{|w^l|})$ where $\overline{|w^l|}$ corresponds to the mean absolute weight value in layer $l$. We run each layer-$\beta$ combination experiment multiple times – to account for fluctuation in the randomised noise – and present the results in Figure 1. Even though the mean variation of noise added is the same, noise relative to the original weight value (multiplicative noise) is much better tolerated than absolute (additive noise). Since moving weights to quantisation centres is analogous to adding noise, we translate these results into our approach and prioritise weights that have small relative distances to be clustered first. We find avoiding significant quantisation errors requires ensuring that $\frac{|\delta w_i|}{|w_i|}$ is small for all weights. If this is not possible, then performance could suffer. In this case, we create an additional cluster centroid in the vicinity of an existing cluster to reduce this relative distance. Our work also challenges the almost universal trend in the literature (Yuhang Li, Xin Dong, 2020; Jung et al., 2019; Zhang et al., 2018a; Zhou et al., 2016; Yamamoto, 2021; Oh et al., 2021) of leaving the first and last layers either at full precision or 8-bit; we attempt a full network quantisation. The cost of not quantising the first layer – which typically requires the most re-use of weights due to the larger resolution of input maps – and the final linear layer – which often contains the largest number of unique weight values – is too significant to ignore.

With multiple stages of training and clustering, we finish with a significantly reduced set of unique weights. The regularisation term encourages high probability regions in the weight distribution and a lower-entropy weight-space. The initial choice of cluster centroids as powers-of-two helps us meet our third objective – energy-saving multiplication. Overall we find four distinct advantages over the works reviewed:

- We assign a cluster value to *all weights* — including the first and last layers.

- We emphasise a low entropy encoding with a regularisation term, achieving entropies smaller than even those seen using 3-bit quantisation approaches – over which we report superior performance.

- We require no additional layerwise scaling; the unique weights are shared across all layers.

- WFN substantially reduces the number of unique parameters in a network when compared to existing SOTA quantisation approaches.

## 2 RELATED WORK

**Clip and Scale Quantisation.** Typical quantisation approaches reduce the number of bits used to represent components of the network. Quantisation has been applied to all parts of the network with varying success; the weights, gradients, and activations have all been attempted (Hubara et al., 2016; Lee et al., 2017; Jung et al., 2019; Yang et al., 2019; Shkolnik et al.; Zhou et al., 2016). Primarily, these approaches are motivated by the need to reduce the energy costs of the multiplication of 32-bit floating-point numbers. This form of quantisation uses a rounding function and scaling factor ($sf$) in making value adjustments where a weight $w_i$ is mapped to a new value $w_i' = sf * round(w_i)$ for some predetermined rounding function. The scaling factor (determined by a clipping range) can be learned channel-wise (Jacob et al., 2018; Zhang et al., 2018c), or more commonly, layerwise with separate formulations. This results in different channels/layers having a diverse pool of mapping values for the network weights/activations/gradients. Quantisation can be performed without training — known as post-training quantisation, or with added training steps – called quantisation-aware training (QAT). Retraining incurs higher initial costs but results in superior performance. A clipping+scaling quantisation example relevant to our own is the work of (Zhou et al., 2017), where the authors restrict the layerwise rounding of weights to powers-of-two. The use of powers-of-two has the additional benefit of energy-cheap bit-shift multiplication. A follow-up piece of work (Yuhang Li, Xin Dong, 2020) suggests additive powers-of-two (APoT) instead to capture the pre-quantised distribution of weights better.

**Weight Sharing Quantisation.** Other formulations of quantisation do not use clipping and scaling factors (Stock et al., 2020; Tartaglione et al., 2021; Wu et al., 2018). Instead, they adopt clustering techniques to cluster the weights and fix the weight values to their assigned group cluster centroid. These weights are stored as codebook indices, allowing for compressed representation methods such as Huffman encoding to squeeze the network further. Unlike clipping+scaling quantisation techniques, and like ours, these methods share the pool of weights across the entire network. The work by (Wu et al., 2018) is of particular interest since both the motivation and approach are related to ours. Here the authors use a *spectrally relaxed* k-means regularisation term to encourage the network weights to be more amenable to clustering. In their case, they focus on a filter-row codebook inspired by the row-stationary dataflow used in some accelerator designs (Chen et al., 2017). However, their formulation is explored only for convolution, and they restrict clustering to groups of weights (filter rows) rather than individual weights due to computational limitations as recalibrating the k-means regularisation term is expensive during training. Similarly (Stock et al., 2020; Fan et al., 2021), focus on quantising groups of weights into single codewords rather than the individual weights themselves. Weight-sharing approaches similar to ours include (Ullrich et al., 2017). The authors use the distance from an evolving Gaussian mixture as a regularisation term to prepare the clustering weights. Although it is successful with small dataset-model combinations, the complex optimisation — particularly the additional requirement for Inverse-Gamma priors to lower-bound the mixture variance to prevent mode collapse— limits the method's practical applicability due to the high computational costs of training. In our formulation, the weights already fixed no longer contribute to the regularisation prior, reducing the computational overhead.

## 3 METHOD

**Quantisation.** Consider a network $\mathcal{N}$ parameterised by $N$ weights $W = \{w_1, ..., w_N\}$. Quantising a network is the process of reformulating $\mathcal{N}' \leftarrow \mathcal{N}$ where the new network $\mathcal{N}'$ contains weights which all take values from a reduced pool of $k$ cluster centres $C = \{c_1, ..., c_k\}$ where $k \ll N$. After quantisation, each of the connection weights in the original network is replaced by one of the cluster centres $w_i \leftarrow c_j$, $W' = \{w_i'|w_i' \in C, i = 1, \cdots, N\}$, $|W'| = k$, where $W'$ is the set of weights of the new network $\mathcal{N}'$, which has the same topology as the original $\mathcal{N}$.

**Method Outline.** WFN is comprised of $T$ *fixing iterations* where each iteration $t \in T$ has a training and a clustering stage. The clustering stage is tasked with partitioning the weights into two subsets $W = W_{fixed}^t \cup W_{free}^t$. $W_{fixed}^t$ is the set of weights fixed at one of the cluster centre values $c_k \in C$. The weights $w_i \in W_f^t ixed$ are not updated by gradient decent in this, nor any subsequent training stages, these weights are *fixed*. In contrast, the *free-weights* denoted by $W_{free}^t$ remain trainable during the next training stage. With each subsequent iteration $t$ we increase the proportion of weights that take on fixed cluster centre values $p_t$ such that $p_0 < p_1 \ldots < p_T$ , where $p_t = \frac{|W_{fixed}^t|}{|W|}$. By iteration $T$, all weights will be fixed to one of the cluster centres making $p_T = 1$. The training stage combines gradient descent based error correction along with a tight-cluster regularisation term and is tasked with maintaining lossless performance as we fix more of the weights to cluster centres.

**Clustering Stage.** In the clustering stage, we work backwards from our goal of minimising the relative distance travelled for each of the weights to determine which values cluster centres $c_i \in C$ should take. Given a weight $w_i \in W$ and cluster centre $c_j \in C$ we use a distance function $D(w_i, c_j) = |w_i - c_j|$ and the original weight value $w_i$ to define a relative distance measure $D_{rel}(w_i, c_j) = \frac{D(w_i, c_j)}{|w_i|}$. Normalising by $w_i$ ensures that distance is measured proportionally to the original weight value, rather than in absolute terms. To use this in determining the cluster centres, we enforce a threshold $\delta$ on this relative distance, $D_{rel}(w_i, c_j) \leq \delta$ for small $\delta$. We can then define the cluster centres $c_j \in C$ which make this possible using a simple recurrence relation. Assume we have a starting cluster centre value $c_j$, we want the neighbouring cluster value $c_{j+1}$ to be such that if a network weight $w_i$ is between these clusters $w_i \in [c_j, \frac{c_{j+1}+c_j}{2}]$ then $D_{rel}(w_i, c_j) \leq \delta$. Plugging in $\frac{c_{j+1}+c_j}{2}$ and $c_j$ into $D_{rel}$ and setting it equal to $\delta$ we have:

$$\frac{|\frac{c_{j+1}+c_j}{2} - c_j|}{\frac{c_{j+1}+c_j}{2}} = \delta, \tag{1}$$

this leads to a recurrence relation:

$$c_{j+1} = c_j\left(\frac{1+\delta}{1-\delta}\right),\ 0 < \delta < 1, \tag{2}$$

that provides the next cluster centre value given the previous one. With this, we can generate all the cluster centres given some boundary conditions. Starting with some value $c_0$, we generate $c_1$ which is then used to generate $c_2$ and so on. We introduce the lower-bound cluster threshold $c_0 = \delta_0$, which we use to generate the cluster centres and any absolute weight value that is less than $\delta_0$ is assigned the cluster value 0. This lower bound serves two purposes: firstly, it reduces the number of proposal cluster centres which would otherwise increase exponentially in density around zero, and additionally, the zero-valued weights will allow sparsity-leveraging hardware to avoid operations that use these weights, reducing the computational overhead. As an upper-bound, we stop the recurrence once a cluster centre is larger than the maximum weight in the network, $\max_j |c_j| \leq \max_i |w_i|,\ w_i \in W, c_j \in C$.

**Generating the Proposed Cluster centres.** Putting this together, we have a starting point $c_0 = \delta_0$, a recurrence relation to produce cluster centres given $c_0$ that maintains a relative distance change when weights are moved to their nearest cluster centre, and a centre generation stopping condition $c_j \leq \max_{i \in W} |w_i|, c_j \in C$. We use the $\delta_0$ value as our first proposed cluster centre $c_0$ with the recurrence relation generating a proposed cluster set of size $s$. Since all these values will contain only positive values, we join this set with its negated version along with a zero value to create a proposal cluster set $C^S = \{a(\frac{1+\delta}{1-\delta})^j \delta_0 \mid j = 0, 1 \cdots s;\ a = +1, 0, -1\}$ of size $2s + 1$.

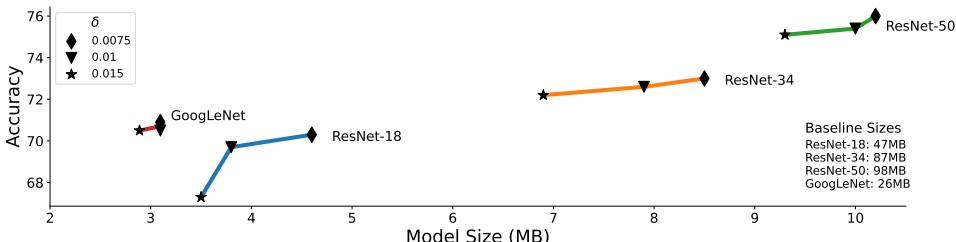

Figure 3: The accuracy vs model size trade-off can be controlled by the $\delta$ parameter. All experiments shown are using the ImageNet dataset, accuracy refers to top-1.

To account for the zero threshold $\delta_0$ and for ease of notation as we advance, we make a slight amendment to the definition of the relative distance function $D_{rel}(w_i, c_j)$ :

$$D_{rel}^+(w_i, c_j) = \begin{cases} \frac{D(w_i, c_j)}{|w_i|}, & \text{if } |w_i| \geq \delta_0 \\ 0, & \text{otherwise} \end{cases} \tag{3}$$

**Reducing $k$ with Additive Powers-of-two Approximations.** Although using all of the values in $C^S$ as centres to cluster the network weights would meet the requirement for the relative movement of weights to their closest cluster to be less than $\delta$, it would also require a large number of $k = |C^S|$ clusters. In addition, the values in $C^S$ are also of full 16-bit precision, and we would prefer many of the weights to be powers-of-two for ease of multiplication in hardware. With the motivation of reducing $k$ and encouraging powers-of-two clusters whilst maintaining the relative distance movement where possible, we look to a many-to-one mapping of the values of $C^S$ to further cluster the cluster centres. Building on the work of others (Zhou et al., 2017; Yuhang Li, Xin Dong, 2020), we map each of the values $c_i \in C^S$ to their nearest power-of-two, $round(c_i) = sgn(c_i)2^{\lfloor log_2(c_i) \rceil}$ and, for flexibility, we further allow for *additive* powers-of-two rounding. With additive powers-of-two rounding, each cluster value can also come from the sum of powers-of-two values ($b$-bit) up to order $\omega$ where the order represents the number of powers-of-two that can contribute to the approximation[2]. As an example, given $c_k = 0.45, c_k \in C^S$, an $\omega = 1$ approximation would round $c_k$ to its nearest power-of-two $c_k^1 = \frac{1}{2^1} = 0.5$ whereas an order $\omega = 2$ approximation is $c_k^2 = \frac{1}{2^1} - \frac{1}{2^4} = 0.4375$. A higher-order grants a better approximation, but at the cost of more cluster centroids overall and requires $\omega$ bit-shifts + additions in hardware. We give full details of the algorithm to map the proposal set $C^S$ to the $\omega$-order approximation $\widetilde{C}^\omega$ in the Appendix.

**Minimalist Clustering.** We are now ready to present the clustering procedure for a particular iteration $t$, which we give the pseudo-code for in Algorithm 1. We start the iteration with $\omega = 1$ and a set of weights not yet fixed $W_{free}^t$. For the set of cluster centres $\widetilde{C}^\omega$ of order $\omega$, let $c_*^\omega(i) = \min_{c \in \widetilde{C}^\omega} D_{rel}^+(w_i, c)$ be the one closest to weight $w_i$. $n_k^\omega = \sum_i \mathbb{I}[c_k^\omega = c_*^\omega(i)]$ counts the number of weights assigned to cluster centre $c_k^\omega \in \widetilde{C}^\omega$, where the indicator function $\mathbb{I}[x]$ is 1 if $x$ is true and 0 otherwise. Let $k^* = \arg\max_k n_k^\omega$ so that $c_{k^*}^\omega$ is the modal cluster. For the cluster $k^*$ let permutation $\pi$ of $\{1, \ldots, N\}$ that maps $w_i \mapsto w'_{\pi(i)}$, be such that the sequence $(w'_1(k^*), w'_2(k^*), \ldots, w'_N(k^*))$ is arranged in ascending order of relative distance from the cluster $c_{k^*}^\omega$. In other words, $D_{rel}^+(w'_i(k^*), c_{k^*}^\omega) \leq D_{rel}^+(w'_{i+1}(k^*), c_{k^*}^\omega)$, for $i = 1, \ldots, (N-1)$. We choose $n$ to be the largest integer such that:

$$\sum_{i=1}^n D_{rel}^+(w'_i(k^*), c_{k^*}^\omega) \leq n\delta, \text{ and } \sum_{i=1}^{n+1} D_{rel}^+(w'_i(k^*), c_{k^*}^\omega) > (n+1)\delta, \tag{4}$$

---

[2]The notion of an order has been identified recently in work parallel to ours (Oh et al., 2021), where the term *word* is used instead, i.e. *two-word* directly translates into an $\omega = 2$.

and define $\{w'_1, w'_2, \ldots, w'_n\}$ to be the set of weights to be fixed at this stage of the iteration. These are the weights that can be moved to the cluster centre $c^{\omega}_{k*}$ without exceeding the average relative distance $\delta$ of the weights from the centre. The corresponding weight indices from the original network $\mathcal{N}$ are in $\{\pi^{-1}(1), \ldots, \pi^{-1}(n)\}$, and called $fixed_{new}$ in the algorithm. If there are no such weights that can be found, *i.e.*, for some cluster centre $l^*$, the minimum relative distance $D^+_{rel}(w'_1(l^*), c_{l*}) > \delta$, the corresponding set $fixed_{new}$ is empty. In this case, there are no weights that can move to this cluster centre without breaking the $\delta$ constraint and we increase order $\omega \leftarrow \omega + 1$ to compute a new $c^{\omega}_{k*}$, repeating the process until $|fixed_{new}| > 0$. Once the condition $|fixed_{new}| > 0$ is satisfied, we fix the identified weights $\{w'_1, w'_2, \ldots, w'_n\}$ to their corresponding cluster centre value $c^{\omega}_{k*}$ and move them into $W^{t+1}_{fixed}$. We continue the process of identifying cluster centres and fixing weights to these centres until $|W^{t+1}_{fix}| \geq Np_t$, at which point the iteration $t$ is complete and the training stage of iteration $t + 1$ begins. Our experiments found that a larger $\delta$ has less impact on task performance during early $t$ iterations and so we reduce $\delta$ linearly with each increasing $t \in T$ (full details in the Appendix). We will show later that, with a small $\delta$, over 75% of the weights can be fixed with $\omega = 1$ and over 95% of weights with $\omega \leq 2$.

**Training Stage.** Despite the steps taken to minimise the impact of the clustering stage, without retraining, performance would suffer. To negate this, we perform gradient descent to adjust the remaining free weights $W^t_{free}$. This allows the weights to correct for any loss increase incurred after clustering where training aims to select values $W^t_{free}$ that minimise the usual loss (cross-entropy in our case) whilst $W_{fixed}$ remain unchanged.

**Cosying up to Clusters.** Having the remaining $W^t_{free}$ weights closer to the cluster centroids $C$ post-training makes clustering less damaging to performance. We coerce this situation by adding a regularisation term — given in Equation 5 — to the retraining where $p(c_j|w_i) = \frac{e^{-D^+_{reg}(w_i, c_j)}}{\sum_l^k e^{-D^+_{reg}(w_i, c_l)}}$. The idea is to penalise the free-weights $W^t_{free}$ proportionally to their distance away from the clusters to which they are most likely to be assigned — i.e. the closest. Clusters that are unlikely to be weight $w_i$'s nearest — and therefore final fixed value — do not contribute much to the penalisation term. We balance this regularisation term against the cross-entropy training loss with the $\gamma$ hyper-parameter, which we fix to be a $\alpha$ proportion of the other loss term $\gamma = \alpha \frac{\mathcal{L}_{cross\_entropy}}{\mathcal{L}_{reg}}$. We note that the $\gamma$ term is detached from the computational graph and treated as a constant (otherwise the $L_{reg}$ term would cancel).

$$\mathcal{L}_{reg} = \gamma \sum_{i \in W_{free}}^N \sum_j^k D^+_{reg}(w_i, c_j) p(c_j|w_i) \tag{5}$$

## 4 EXPERIMENT DETAILS

We apply WFN to fully converged models trained on the CIFAR-10 and ImageNet datasets. Our pre-trained models are all publicly available with strong baseline accuracies[3]: Resnet-(18,34,50) (Wu et al., 2017), MobileNetV2 (Sandler et al., 2018) and, GoogLeNet (Chollet, 2017). We run ten weight-fixing iterations for three epochs, increasing the percentage of weights fixed until all weights are fixed to a cluster. In total, we train for 30 epochs per experiment using the Adam optimiser (Kingma & Ba, 2015) with a learning rate $2 \times 10^{-5}$. We use grid-search to explore hyper-parameter combinations using ResNet-18 and MobileNetV2 models with the CIFAR-10 dataset and find that the regularisation weighting $\alpha = 0.4$ works well across all experiments reducing the need for further hyper-parameter tuning as we advance. The distance threshold $\delta$ gives the practitioner control over the compression-performance trade-off (see Figure 3), and so we report a range of values. We give full details of the experiments along with the results of a hyper-parameter ablation study using CIFAR-10 in the Appendix.

---

[3]CIFAR-10 models : https://github.com/kuangliu/pytorch-cifar, ImageNet models: torchvision

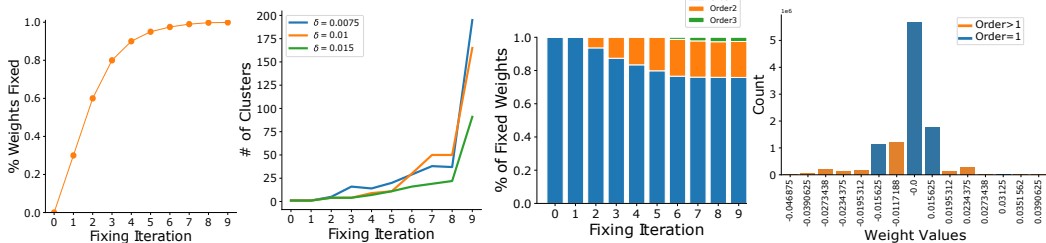

Figure 4: **Far left:** We increase the number of weights in the network that are fixed to cluster centres with each fixing iteration. **Middle left:** Here, we show how decreasing the $\delta$ threshold increases the number of cluster centres, but only towards the last few fixing iterations, which helps keep the weight-space entropy down. **Middle right:** The majority of all weights are order 1 (powers-of-two), the increase in order is only needed for outlier weights in the final few fixing iterations. **Far right:** The weight distribution (top-15 most used show) is concentrated around just four values. All of these charts are produced with ResNet-18 models trained on the ImageNet dataset.

## 5 RESULTS

| Model | Method | Accuracy (%) | | | Model | Method | Accuracy (%) | | |
|---|---|---|---|---|---|---|---|---|---|
| | | Top-1 | Top-5 | CR | | | Top-1 | Top-5 | CR |
| ResNet-18 | Baseline | 68.9 | 88.9 | 1.0 | ResNet-34 | Baseline | 73.3 | 90.9 | 1.0 |
| | LQ-Net | 68.2 | 87.9 | 7.7 | | LQ-Net | 71.9 | 90.2 | 8.6 |
| | APoT | 69.9 | 89.2 | 10.2 | | APoT | 73.4 | 91.1 | 10.6 |
| | LSQ | 70.2[+] | 89.4[+] | 9.0[*] | | LSQ | 73.4[+] | 91.4[+] | 9.2[*] |
| | WFN ($\delta = 0.015$) | 67.3 | 87.6 | 13.4 | | WFN($\delta = 0.015$) | 72.2 | 90.9 | 12.6 |
| | WFN ($\delta = 0.01$) | 69.7 | 89.2 | 12.3 | | WFN ($\delta = 0.01$) | 72.6 | 91.0 | 11.1 |
| | WFN ($\delta = 0.0075$) | 70.3 | 89.1 | 10.2 | | WFN ($\delta = 0.0075$) | 73.0 | 91.2 | 10.3 |
| ResNet-50 | Baseline | 76.1 | 92.8 | 1.0 | GoogLeNet | Baseline | 69.7 | 89.6 | 1.0 |
| | LQ-Net | 74.2 | 91.6 | 5.9 | | Deep $k$-Means | 69.4 | 89.7 | 3.0 |
| | APoT | 75.8 | 92.7 | 9.0 | | GreBdec | 67.3 | 88.9 | 4.5 |
| | LSQ | 75.8[+] | 92.7[+] | 8.1[*] | | KQ | 69.2 | - | 5.8 |
| | WFN ($\delta = 0.015$) | 75.1 | 92.1 | 10.3 | | WFN($\delta = 0.015$) | 70.5 | 89.9 | 9.0 |
| | WFN ($\delta = 0.01$) | 75.4 | 92.5 | 9.8 | | WFN ($\delta = 0.01$) | 70.5 | 90.0 | 8.4 |
| | WFN ($\delta = 0.0075$) | 76.0 | 92.7 | 9.5 | | WFN ($\delta = 0.0075$) | 70.9 | 90.2 | 8.4 |

Table 1: A comparison of WFN against other quantisation and weight clustering approaches. The WFN pipeline is able to achieve higher compression ratios than the works compared whilst matching or improving upon baseline accuracies.

[*] Estimated from the LSQ paper model size comparison graph, we over-estimate to be as fair as possible.

[+] Open-source implementations have so far been unable to replicated the reported results: https://github.com/hustzxd/LSQuantization.

We begin by comparing WFN for a range of $\delta$ values against a diverse set of quantisation approaches that have comparable compression ratios (CR) in Table 5. The 3-bit quantisation methods we compare include: LSQ (Esser et al., 2020), LQ-Net (Zhang et al., 2018b) and APoT (Yuhang Li, Xin Dong, 2020). We additionally compare with the clustering-quantisation methods using the GoogLeNet model: Deep-$k$-Means (Wu et al., 2018) whose method is similar to ours, KQ (Yu et al., 2020), and GreBdec (Yu et al., 2017). Whilst the results demonstrate WFN's lossless performance with SOTA CR, this is not the main motivation for the method. Instead, we are interested in how WFN can reduce the number of unique parameters in a network and corresponding weight-space entropy which together give the potential benefit of future accelerator designs to reduce data movement costs. We note to the interested reader there are additional theoretical reasons for pursuing low weight-space entropy networks (Hinton & van Camp, 1993; Grünwald, 2000; Hansen & Yu, 2001), but our primary focus is from the perspective of reducing accelerator data movement costs. Capturing the benefit of algorithmic developments to reduce data movement costs in accelerator designs is non-trivial due to the significant dependency on architecture memory hierarchy, communication orchestration costs and the dataflow mapping used (Sze et al., 2020; Kwon et al., 2019; Sakr et al., 2017). Some active works attempt to provide robust estimations in bridging the gap between estimation and actual energy costs. This estimates energy efficiencies for various accelerator mappings (Kwon et al., 2019), or fixes the mapping and estimates the energy costs on a particular accelerator (Yang et al., 2017). However, this support is limited to specific fixed architecture designs and does not account for the referenced weight-sharing that we introduce. Instead, we rely on analytical model metrics such as the one proposed by (Wu et al., 2018), itself a subset of measurements con-

| Model | Method | Top-1 | Entropy | Param Count | $\mathrm{Rep}(\mathcal{N}')$ | Model Size |
|---|---|---|---|---|---|---|
| ResNet-18 | Baseline | 68.9 | 23.3 | 10756029 | 1.000 | 46.8MB |
| | APoT (3bit) | 69.9 | 5.77 | 9237 | 0.283 | 4.56MB |
| | WFN ($\delta = 0.015$) | 67.3 | 2.72 | 90 | 0.005 | 3.5MB |
| | WFN ($\delta = 0.01$) | 69.7 | 3.01 | 164 | 0.007 | 3.8MB |
| | WFN ($\delta = 0.0075$) | 70.3 | 4.15 | 193 | 0.018 | 4.6MB |
| ResNet-34 | Baseline | 73.3 | 24.1 | 19014310 | 1.000 | 87.4MB |
| | APoT (3bit) | 73.4 | 6.77 | 16748 | 0.296 | 8.23MB |
| | WFN ($\delta = 0.015$) | 72.2 | 2.83 | 117 | 0.002 | 6.9MB |
| | WFN ($\delta = 0.01$) | 72.6 | 3.48 | 164 | 0.002 | 7.9MB |
| | WFN ($\delta = 0.0075$) | 73.0 | 3.87 | 233 | 0.004 | 8.5MB |
| ResNet-50 | Baseline | 76.1 | 24.2 | 19915744 | 1.000 | 97.5MB |
| | WFN ($\delta = 0.015$) | 75.1 | 3.55 | 125 | 0.002 | 9.3MB |
| | WFN ($\delta = 0.01$) | 75.4 | 4.00 | 199 | 0.002 | 10.0MB |
| | WFN ($\delta = 0.0075$) | 76.0 | 4.11 | 261 | 0.003 | 10.2MB |

Table 2: A full metric comparison of WFN Vs. APoT. For readiblity $\mathrm{Rep}(\mathcal{N}')$ is given relative to baseline. Note, APoT did not release the ResNet-50 model save which is why it is not compared here.

ducted by (Sakr et al., 2017), in terms of the number $N_w$ of operations that each of the $|W'|$ unique weights of bit-width $B_w$ is involved in. The low-entropy encoding that we aspire to is motivated by the accelerator designs (Moons & Verhelst, 2016; Han et al., 2016) that seeks to exploit the Huffman encoding possible for the network index set. Hence, the index for each weight $w_i \in W'$ can be represented with a bit-width of $B_{w_i}$ and we can account for the number $N_{w_i}$ of times they are used in an inference computation. Thus, instead of the representational cost $N_w|W'|B_w$ of (Sakr et al., 2017) for the final network $\mathcal{N}'$, we use as metric:

$$\mathrm{Rep}(\mathcal{N}') = \sum_{w_i \in W'} N_{w_i} B_{w_i} \tag{6}$$

The authors of the APoT have released the quantised model weights and code, and we use the released model-saves[4] of this SOTA model to compare the entropy, $\mathrm{Rep}(\mathcal{N}')$, unique parameter count, model size and accuracy in Table 2. Our work outperforms APoT in weight-space entropy, unique parameter count and weight representational cost by a large margin. Taking the ResNet-18 experiments as an example, the reduction to just 164 weights compared with APoT's 9237 demonstrates the effectiveness of WFN. This is in part due to our full-network quantisation (APoT, as aforementioned, does not quantise the first, last and batch-norm parameters), but even when we discount these advantages and look at weight subsets ignoring the first, last and batch-norm layers WFN uses many times fewer parameters and half the weight-space entropy. We provide a further breakdown of these results in the Appendix (Table 4). Finally, let us examine how WFN achieves the substantial reduction in weight-space entropy. In Figure 4 we see that not only do the resultant WFN networks have very few unique weights, but the weight distribution is such that the vast majority of all of the weights are a small handful of powers-of-two values (order 1) and the other unique weights (outside of the top 4) are of low frequency and added only in the final fixing iterations.

## 6   CONCLUSION

We have presented WFN, a compression pipeline that can successfully compress whole neural networks. A single network codebook, focusing on reducing entropy of the weight-space coupled with the number of unique weights in the network. The WFN process produces highly compressible and potentially hardware-friendly representations of networks using just a few unique weights without performance degradation. Although the results demonstrate strong compression ability, few unique parameters, and low weight-space entropy, the true potential of the method is that it will gives accelerator designers more scope for weight re-use, keeping most/all weights close to computation, reducing the energy-hungry data movement costs.

---

[4]https://github.com/yhhhli/APoT_Quantization

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

# A APPENDIX

## A.1 THE WFN OVERVIEW ALGORITHM

---

**Algorithm 1:** Clustering $Np_t$ weights at the $t^{th}$ iteration.

---

1 **while** $|W_{fix}^{t+1}| \leq Np_t$ **do**
2      $\omega \leftarrow 0$
3      $fixed_{new} \leftarrow [\,]$
4      **while** $fixed_{new}$ *is empty* **do**
5          Increase the order $\omega \leftarrow \omega + 1$
6          List the cluster centres in $\widetilde{C}^\omega$ with smallest distances to each $w_i \in W_{free}^{t+1}$
7          Set the cluster centre with the most weights assigned as $c_*^\omega(i)$
8          Sort the weights $w_i \in W_{free}^{t+1}$ by their distance to $c_*^\omega(i)$
9          **do**
10              Append to $fixed_{new}$ the $w_i \in W_{free}^{t+1}$ with next smallest distance
11          **while** *The mean* $D_{rel}^+(w \in fixed_{new}, c_*^\omega(i)) \leq \delta$
12      Fix the all the weights in $fixed_{new}$ to cluster centre $c_*^\omega(i)$

---

## A.2 DETAILS OF THE POWERS-OF-TWO APPROXIMATION ALGORITHM

We map our proposal set $C^S$ to a $\omega$-order approximation where each of the clusters $c_k \in C^S$ are written as $\omega$ powers-of-two (Eq 7). We do so using Algorithm 2. Figure 5 demonstrates how the values of $C^S$ are rounded given different orders.

$$c_k = \sum_{j=1}^{\omega} r_j, \ r_j \in \{-\frac{1}{2^b}, \ldots, -\frac{1}{2^{j+1}}, -\frac{1}{2^j}, 0, \frac{1}{2^j}, \frac{1}{2^{j+1}}, \ldots \frac{1}{2^b}\} \tag{7}$$

---

**Algorithm 2:** Determining possible clusters

---

1 **Input:** The full precision proposal set: $C^S$, allowable relative distance: $\delta$, pow2 rounding
     function: $round(x) = sgn(x)2^{\lfloor log_2(x) \rceil}$
2 **Output:** An order $\omega$ precision cluster set: $\widetilde{C}^\omega$
3 $\widetilde{C}^\omega \leftarrow [\,]$
4 **for** $c_k \in C^S$ **do**
5      $c_k' = round(c_k)$
6      **for** $i = 0 \rightarrow \omega$ **do**
7          $\delta_{c_k} \leftarrow c_k - c_k'$
8          **if** $|\delta_{c_k}| \geq \delta c_k$ **then**
9              $c_k' \leftarrow c_k' + round(\delta_{c_k})$
10          **end**
11      **end**
12      $\widetilde{C}^\omega \leftarrow \widetilde{C}^\omega \cup \{c_k'\}$
13 **end**

---

## A.3 REDUCING $\delta$

To account for the observation that smaller relative distances are well tolerated in the early stages but less so as the final weights are quantised (high $t$), $\delta$ linearly. For any given $t \in T$ we use $\delta^t = \delta(T - t + 1)$ as our threshold.

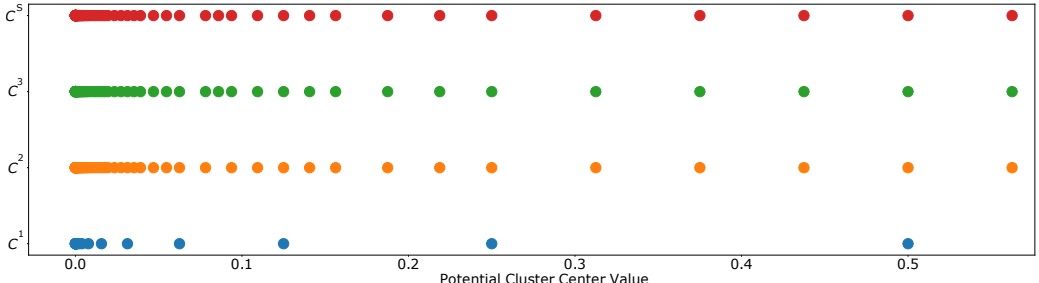

Figure 5: Approximating clusters in $C^S$ with different orders

## A.4 EXPERIMENT DETAILS

We give a full breakdown of the parameters used across all experiments ran in Table 3.

| Model | Data | Opt | LR | $T$ | Epochs per T | Batch size | $\gamma$ | $\alpha$ |
|-------|------|-----|-----|-----|-------------|-----------|---------|---------|
| ResNet-18 | ImageNet | Adam | 2e-4 | 10 | 3 | 128 | $\{0.05, 0.025, , 0.015, 0.01, 0.0075, 0.005\}$ | $\{0.2, 0.4\}$ |
| ResNet-34 | ImageNet | Adam | 2e-4 | 10 | 3 | 64 | $\{0.05, 0.025, , 0.015, 0.01, 0.0075, 0.005\}$ | $\{0.4\}$ |
| ResNet-50 | ImageNet | Adam | 2e-4 | 10 | 3 | 64 | $\{0.05, 0.025, , 0.015, 0.01, 0.0075, 0.005\}$ | $\{0.4\}$ |
| GoogLeNet | ImageNet | Adam | 2e-4 | 10 | 3 | 64 | $\{0.01, 0.0075, 0.015\}$ | $\{0.4\}$ |
| ResNet-18 | CIFAR-10 | Adam | 3e-4 | 10 | $\{3, 5, 10\}$ | 512 | $\{0.01, 0.02, 0.03, 0.04, 0.05\}$ | $\{0.0, 0.1, 0.2, 0.4, 0.8\}$ |
| MobileNet | CIFAR-10 | Adam | 2e-4 | 10 | $\{3, 5, 10\}$ | 512 | $\{0.01, 0.02, 0.03, 0.04, 0.05\}$ | $\{0.0, 0.1, 0.2, 0.4, 0.8\}$ |

Table 3: Full set of hyper-parameters explored for each model-dataset combination.

## A.5 HYPER-PARAMETER EXPLORATION

We conducted a hyper-parameter search on both the ResNet-18 and MobileNet (Figures 6, 7). We find similar results across models when selecting hyper-parameters, and additionally, we observe that as we increase the regularisation term weighting, a weight-space entropy reduction. Further, we found that negating this term altogether $\alpha = 0$ has the surprising effect of increasing the resultant entropy values and causes an accuracy performance drop, demonstrating the utility of the regularisation term.

## A.6 LAYERWISE BREAKDOWN

In Figure 8 we examine how the parameter count and layer-parameter entropy change with each layer for both the WFN and APoT approaches. We find both gains over the unquantised layers of APoT, but also that the entropy and parameter count in the convolutional layers (those quantised by APoT) are similar.

## A.7 A FULL METRIC COMPARISON

In Table 4 we give the full metric breakdown comparing WFN to the state-of-the-art APoT work. We calculate the unique parameter count and entropy values for subsets of the weights. No BN corresponds to all weights other than those in the batch-norm layers, and No BN-FL is the set of weights not including the first-last and batch-norm layers. It's clear here that WFN outperforms APoT even when we discount the advantage gained of taking on the challenge of quantising all layers.

## A.8 PRUNING EXPERIMENTS

To explore how WFN interacts with pruning we conduct a simple set of experiments. Instead of starting the WFN process with all weights un-fixed we randomly select $p\%$ of the weights to be pruned in each layer. We then run WFN as before starting with $p_t = p$, reducing the number of $T$ iterations. The results, shown in Figure 9, are conducted with a ResNet-18 and CIFAR-10

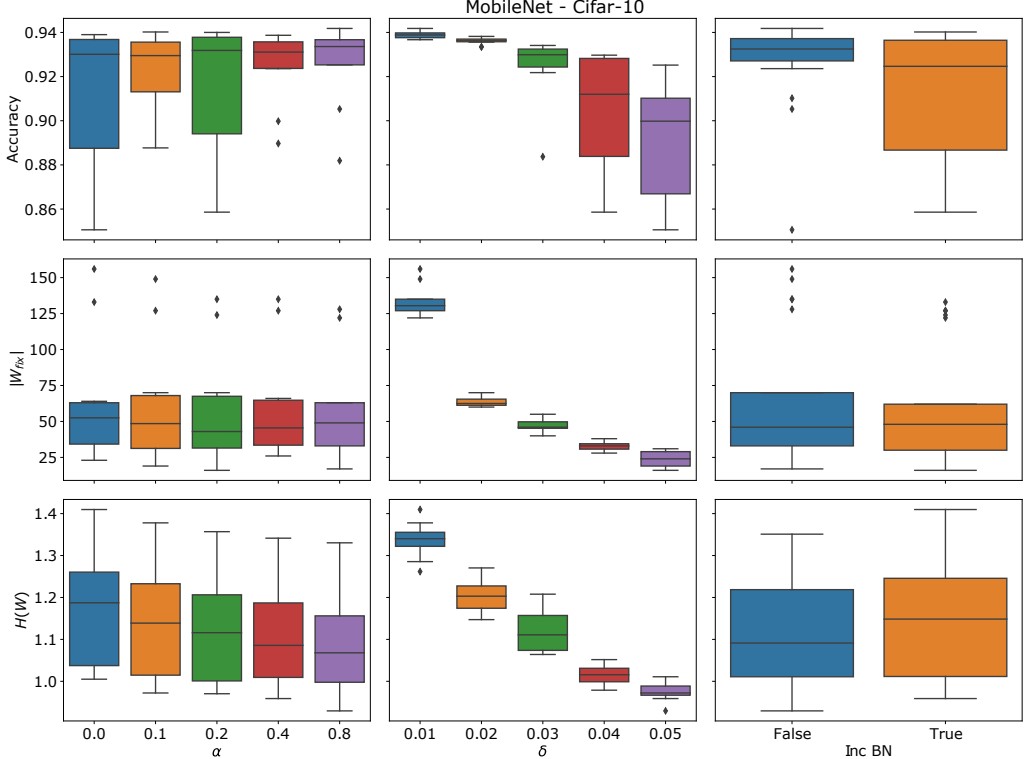

Figure 6: Experiments exploring the hyper-parameter space with MobileNet model trained on the CIFAR-10 dataset. Columns; **Left:** varying the regularisation ratio $\alpha$, middle: varying the distance change value $\delta$, **Right:** whether we fix the batch-norm variables. Rows; top: top-1 accuracy test-set CIFAR-10, **Middle:** total number of network weights, bottom: entropy of the weights.

| Model | Method | Top-1 | Full Network Entropy | Full Network Param Count | No BN Entropy | No BN Param Count | No BN-FL Entropy | No BN-FL Param Count | Model Size |
|---|---|---|---|---|---|---|---|---|---|
| ResNet-18 | Baseline | 68.9 | 23.3 | 10756029 | 23.3 | 10748288 | 23.3 | 10276369 | 46.8MB |
| | APoT (3bit) | 69.9 | 5.77 | 9237 | 5.76 | 1430 | 5.47 | 274 | 4.56MB |
| | WFN ($\delta = 0.015$) | 67.3 | 2.72 | 90 | 2.71 | 81 | 2.5 | 81 | 3.5MB |
| | WFN ($\delta = 0.01$) | 69.7 | 3.01 | 164 | 3.00 | 153 | 2.75 | 142 | 3.8MB |
| | WFN ($\delta = 0.0075$) | 70.3 | 4.15 | 193 | 4.13 | 176 | 3.98 | 162 | 4.6MB |
| ResNet-34 | Baseline | 73.3 | 24.1 | 19014310 | 24.1 | 18999320 | 24.10 | 18551634 | 87.4MB |
| | APoT (3bit) | 73.4 | 6.77 | 16748 | 6.75 | 16474 | 6.62 | 389 | 8.23MB |
| | WFN ($\delta = 0.015$) | 72.2 | 2.83 | 117 | 2.81 | 100 | 2.68 | 100 | 6.9MB |
| | WFN ($\delta = 0.01$) | 72.6 | 3.48 | 164 | 3.47 | 132 | 3.35 | 130 | 7.9MB |
| | WFN ($\delta = 0.0075$) | 73.0 | 3.87 | 233 | 3.85 | 191 | 3.74 | 187 | 8.5MB |
| ResNet-50 | Baseline | 76.1 | 24.2 | 19915744 | 24.2 | 19872598 | 24.20 | 18255490 | 97.5MB |
| | WFN ($\delta = 0.015$) | 75.1 | 3.55 | 125 | 3.50 | 105 | 3.42 | 102 | 9.3MB |
| | WFN ($\delta = 0.01$) | 75.4 | 4.00 | 199 | 3.97 | 169 | 3.88 | 163 | 10.0MB |
| | WFN ($\delta = 0.0075$) | 76.0 | 4.11 | 261 | 4.09 | 223 | 4.00 | 217 | 10.2MB |

Table 4: A full metric comparison of WFN Vs. APoT. We compare the unique parameter count and entropy of all parameters in the full network, as well as the same measures but not including the batch-norm layers (No BN) and the parameters not including the batch-norm and first and last layers (No BN-FL).

combination, painting a mixed picture. On the one hand, WFN and pruning at lower levels ($< 50\%$) is well tolerated and provide two benefits, a lower weight-space entropy and fewer weight-fixing iterations. On the other hand, full-precision networks can tolerate much higher ranges of pruning so there it would seem that a certain amount of synergy between the two approaches is present but this is tempered compared to full precision networks.

It's important to note that WFN already has a form of pruning built-in with the $\delta_0$ value balancing the emphasis on pruning over quantisation.

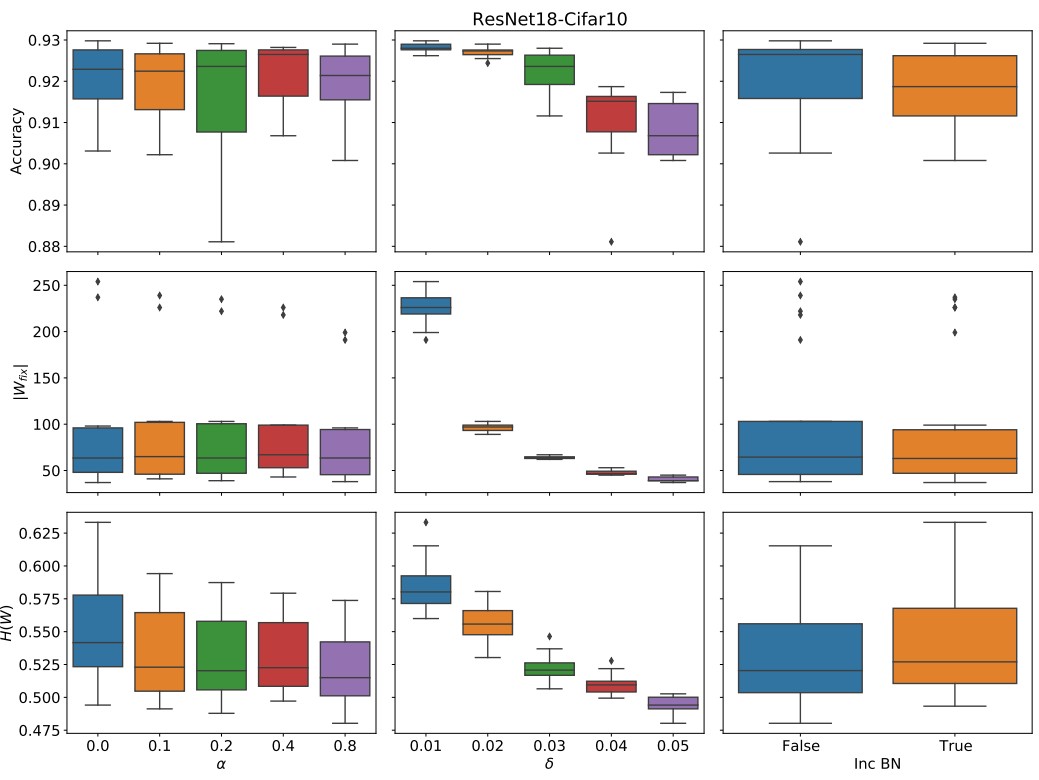

Figure 7: Experiments exploring the hyper-parameter space with ResNet18 model trained on the CIFAR-10 dataset. Columns; **Left:** varying the regularisation ratio $\alpha$, **Middle:** varying the distance change value $\delta$, **Right:** whether we fix the batch-norm variables. Rows; **Top:** top-1 accuracy test-set CIFAR-10, middle: total number of network weights, bottom: entropy of the weights.

Figure 8: We compare WFN with a traditional quantisation set-up (APoT) with varying bit-widths applied to a ResNet18 model trained on the ImageNet dataset. The top chart shows the layerwise unique parameter count where WFN has consistently fewer unique parameters per layer.

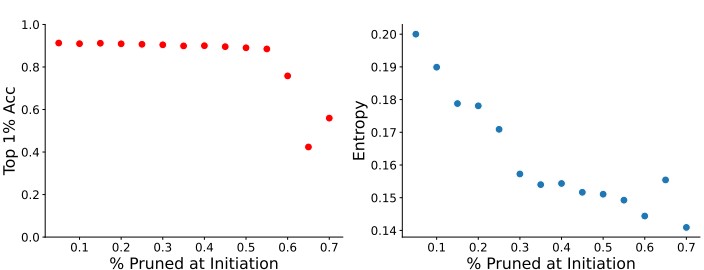

Figure 9: How does WFN and random pruning at initialisation interact?

