# OpenReview forum: "Low Entropy Deep Networks"
_ICLR.cc/2022/Conference — ICLR 2022 Submitted_

### Official Review · Reviewer_vtcG · 2021-10-25

**Correctness:** 3
**Technical Novelty And Significance:** 3
**Empirical Novelty And Significance:** 2
**Recommendation:** 5
**Confidence:** 4

**Main Review:**

Strenths: This paper discusses from the viewpoint of FLOPs do not accurately reflect the complexity of a model, but the data movement costs do. This viewpoint is quite novel to the reviewer and may shed lights into many related works. The technique part is clear and easy to follow. Empirical studies support that this method is able to reduce the parameter counts significantly.

weakness:  The empirical studies seem to be a little bit strange because parameter counts are also a type of "weak proxy" for the energy consumption. How about the reduction of the energy consumption by using WFN? Despite of the parameter counts, what will WFN do to memery and latency? These might be more intereting and important to see than the parameter counts. Because parameter count is not linearly proportion to the latency and memery.

**Summary Of The Paper:**

This paper focuses on reducing the data movement costs by reducing the number of unique weights in a network, i.e., resuing weights. Based on this consideration, a new compression pipeline that can successfully compress whole neural networks has been proposed, called WFN. WFN is based on traditional quantisation manner but aims to cluster the unique weights and reduce the weight-space entropy of the whole network. By doing so, the empirical studies show that the proposed WFN is efficient in terms of reducing parameter counts.

**Summary Of The Review:**

The idea of this paper is novel and may inspire broadly future works. The technique design is clear and correct. The empirical design can be improved.

---

### Official Review · Reviewer_kvqy · 2021-10-31

**Correctness:** 2
**Technical Novelty And Significance:** 3
**Empirical Novelty And Significance:** 2
**Recommendation:** 5
**Confidence:** 3

**Main Review:**

The work is interesting and appears to achieve its goals. Nevertheless, the impact of the work is currently difficult to assess.

A comparison is made between:

a) A scheme that minimises the number of unique weights a network uses:
    - enabling these to be stored locally in each PE
    - the weights are shared across all layers of the network
      (and require no layerwise scaling)
    - weights are then stored as indexes into this (limited) set of weight values

b) Quantizing and storing narrow weights:
     - Quantization is performed and the weight values are stored, e.g. as 3-bit values
     - To support such a scheme additional quantization parameters must be stored
       on a per layer/filter basis and as such increases the number of unique parameters
       in the network.

The argument is made that the approach presented would reduce data movement. What is a realistic expectation here? If we look at the "No BN-FL" param counts it would suggest we would require 81-162 parameters vs. 274. Compared to data movement in general, updating a small numbers of parameters within a PE's memory would seem insignificant? Could a stronger argument be made regarding the reductions in data movement that might result from this work? Infrequent data movements, between periods of significant computation, are of less concern perhaps?

An explanation of precisely what values need to be stored (and how the unique parameter counts are generated) would also make the paper easier to read.

In the case of WFN, how are weights stored precisely? i.e. to determine the model size.

Are the weights in your scheme compressible with a simple scheme (i.e. suitable to be placed on-chip in each PE) in a way that would further reduce data movement? Again, perhaps the potential gains in the reduction of data movement on-chip could be made clearer?


**Summary Of The Paper:**

A technique is developed to reduce the number of unique weights used in a network. The ultimate goal is to exploit this to reduce data movement in hardware. The unique weights are discovered multiple stages of training and clustering. An additional goal is to produce compressible weights. Results demonstrate that good compression ratios are possible, comparable to SoA quantization techniques, but with far fewer numbers of unique parameters.



**Summary Of The Review:**

The work appears interesting but I am unsure about the potential impact on data movement. I found the paper quite difficult to follow in general. For this reason I am currently recommended that the paper is not accepted.

---

### Official Review · Reviewer_geXX · 2021-11-02

**Correctness:** 4
**Technical Novelty And Significance:** 3
**Empirical Novelty And Significance:** 3
**Recommendation:** 8
**Confidence:** 4

**Main Review:**

The most expensive energy costs lie in memory reads. Thus, data read and movement operations, not arithmetic operations, dominate the energy costs of deep learning inference. This work focuses on reducing data movement costs by reducing the number of unique weights in a neural network.

WFN is an interative scheme, where at each iteration there is a training and a clustering stage. The clustering stage carefully choses the closest fixed centers for some of the weights. A method is given to produce and increasingly larger set of cluster centers given some boundary conditions. Further representational reduction is realized by using power of two approximations, which lend themselves to very efficient multiplication by shifting. After each clustering stage, a retraining is performed in order to maintain good performance.

Experimental evaluation on CIFAR-10 and ImageNet datasets shows the compression power of WFN. WFN networks have very few unique weights, and the weight distribution is a small handful of powers-of-two values, while the other unique weights are of low frequency and added only in the final fixing iterations.


**Summary Of The Paper:**

The paper proposes a method called Weight Fixing Networks (WFN), which is designed to minimize the data movement in deep learning inference, thus minimizing the energy cost. WFN aims to maintain very few unique weights, low-entropy weight encodings, unique weight values which are amenable to energy-saving versions of hardware multiplication, while keeping the same performance. A new clustering algorithm for weight generation is shown and analyzed, and experimental evaluation shows the merits of the new scheme.

**Summary Of The Review:**

The paper introduces WFN, weight fixing networks, an algorithm that transforms a neural network into a representation with very few unique weights, the ones most frequently used being powers of two, all while maintaining good inference performance.

---

### Official Review · Reviewer_wcJb · 2021-11-04

**Correctness:** 3
**Technical Novelty And Significance:** 3
**Empirical Novelty And Significance:** 3
**Recommendation:** 5
**Confidence:** 3

**Main Review:**

The paper is well-written (with some minor grammatical errors at some places) and the results shown by the author are comprehensive.
One major weakness I found in the paper is that the analysis is not exactly correct. The authors have shown all experiemntal benefits with parameter counts. But reduction in parameters does not necessarily mean energy efficiency. There is a lot of communication related latency costs that add to energy. The type of router/NoC, their data width etc. play a role in characterizing this. Also, just reducing parameters or unique weights does not mean you will have to access the memory less. Depending upon the PE array size, FIFO width etc. , your data access costs may vary.


Minor comments for the authors are:
1) Is their approach only advantageous for accelerators where weights have to be moved from main memory to PEs? Can the authors comment on how their approach is amenable to many emerging accelerators with weight stationary like flow [1, 2], such as compute-in memory or crossbar based where all weights are already programmed in the hardware? [2] talks about a compute in memory architecture for quantized neural networks which is relevant to the author's proposition.
[1]Shafiee, Ali, et al. "ISAAC: A convolutional neural network accelerator with in-situ analog arithmetic in crossbars." ACM SIGARCH Computer Architecture News 44.3 (2016): 14-26.
[2] Vasquez, Karina, et al. "Activation Density based Mixed-Precision Quantization for Energy Efficient Neural Networks." arXiv preprint arXiv:2101.04354 (2021).
2) The authors have shown that their work performs better than many clip and scale quantization works. Can the authors comment if their work has any resemblance to pruning methods? For e.g. if we start with a pruning at initialization based network and then try to perform the WFN technique), will it have any repurcussions in the clustering and encoding of unique weights.
3) Further, there has been recent works [3, 4] that have tied adversarial attacks with weight sharing/quantization together. Can the authors comment on impliction of WFN and how introducing more unique weights can impact adversarial robustness?
[3]Lin, Ji, Chuang Gan, and Song Han. "Defensive quantization: When efficiency meets robustness." arXiv preprint arXiv:1904.08444 (2019).
[4]Panda, Priyadarshini. "QUANOS: adversarial noise sensitivity driven hybrid quantization of neural networks." Proceedings of the ACM/IEEE International Symposium on Low Power Electronics and Design. 2020.



**Summary Of The Paper:**

This work presents WFN, an ensemble of techniques to reduce data movement costs by reducing the number of unique
weights in a network.  Overall the authors find some distinct advantages in their approach:
They emphasise a low entropy encoding with a regularisation term, achieving entropies smaller than even those seen using 3-bit quantisation approaches. They require no additional layerwise scaling; the unique weights are shared across all layers. They substantially reduces the number of unique parameters in a network when compared to existing SOTA quantisation approaches.


**Summary Of The Review:**

I think the authors have come up with a very interesting technique of methods and are targeting a novel issue of data storage for efficient implementation of neural networks. This paper falls under the same category as pruning or quantization, but provides a very fresh perspective. However, their metrics of evaluation using parameter reduction is weak and does not really support the energy efficiency claim that the authors make. Thus, I have given a rating of 5 for this paper.

---

### Decision · Program_Chairs · 2022-01-20

**Decision:**

Reject

**Comment:**

This paper introduces a method to reduce the number of unique weights in a network. The motivation is that this reduces energy consumption, and can lead to speed-ups.
The reviewers agree that the paper is generally well-written. They also agree that the method presented is interesting and useful to reduce the number of unique weights of the network.
However, reviewers challenge the authors claim that fewer unique weights necessarily lead to lower energy consumption. Operations involved with this method may even lead to more memory transfers. The experimental section supports well the claim that the number of unique weights is decreased at very little cost in terms of accuracy, but does not provide much data in terms of actual energy consumption.
Generally, the reviewers agree that the fundamental idea of this paper is good and should be presented, but since this paper motivation is entirely focused on energy consumption, I suggest that the authors revise the paper and submit it in a future venue. I can see two possible ways to revise the paper: 1) Add actual energy consumption data to the experimental section; or 2) change the motivation of the paper to solely focus on reducing the number of unique weights. It may be useful for some niche applications, or as a starting point for future works.